# Turbulent mass transfer caused by vortex induced reconnection in collisionless magnetospheric plasmas

T.K.M. Nakamura[1], H. Hasegawa [2], W. Daughton[3], S. Eriksson[4], W.Y. Li[5,6] & R. Nakamura[1]

Magnetic reconnection is believed to be the main driver to transport solar wind into the Earth's magnetosphere when the magnetopause features a large magnetic shear. However, even when the magnetic shear is too small for spontaneous reconnection, the Kelvin–Helmholtz instability driven by a super-Alfvénic velocity shear is expected to facilitate the transport. Although previous kinetic simulations have demonstrated that the non-linear vortex flows from the Kelvin–Helmholtz instability gives rise to vortex-induced reconnection and resulting plasma transport, the system sizes of these simulations were too small to allow the reconnection to evolve much beyond the electron scale as recently observed by the Magnetospheric Multiscale (MMS) spacecraft. Here, based on a large-scale kinetic simulation and its comparison with MMS observations, we show for the first time that ion-scale jets from vortex-induced reconnection rapidly decay through self-generated turbulence, leading to a mass transfer rate nearly one order higher than previous expectations for the Kelvin–Helmholtz instability.

[1] Space Research Institute, Austrian Academy of Sciences, 8010 Graz, Austria. [2] Institute of Space and Astronautical Science, Japan Aerospace Exploration Agency, Sagamihara 252-5210, Japan. [3] Los Alamos National Laboratory, Los Alamos, NM 87545, USA. [4] Laboratory for Atmospheric and Space Physics, University of Colorado Boulder, Boulder, CO 80303, USA. [5] Swedish Institute of Space Physics, SE751-21 Uppsala, Sweden. [6] State Key Laboratory of Space Weather, National Space Science Center, Chinese Academy of Sciences, Beijing 100190, China. Correspondence and requests for materials should be addressed to T.K.M.N. (email: takuma.nakamura@oeaw.ac.at)

The Earth's magnetopause, across which the shocked solar wind (magnetosheath) particles are transported into the magnetosphere, consists of velocity and magnetic shears both of which coexist in many boundaries in natural magnetized collisionless plasmas. There are two dominant mechanisms, which transfer mass across such collisionless boundaries. When the boundaries have large magnetic shears, the dominant process is magnetic reconnection which causes very efficient transfer along the reconnected field lines[1–4], while in the limit of super-Alfvénic velocity shear, the Kelvin–Helmholtz instability (KHI)[5–7] is also believed to induce a considerable transport[8–14]. When considering a density asymmetry across the velocity shear, the unstable condition for the KHI is written as follows[15]:

$$\gamma_{KH}^2 = \frac{\rho_1 \rho_2}{(\rho_1 + \rho_2)^2} [\mathbf{k} \cdot (\mathbf{U}_1 - \mathbf{U}_2)]^2$$
$$- \frac{1}{\mu_0(\rho_1 + \rho_2)} \left[ (\mathbf{k} \cdot \mathbf{B}_1)^2 + (\mathbf{k} \cdot \mathbf{B}_2)^2 \right] > 0,$$

(1)

where $\rho_s$ ($s = 1, 2$), $\mathbf{U}_s$ and $\mathbf{B}_s$ are mass density, bulk velocity, and magnetic field at each side across the boundary, respectively. Equation (1) predicts that when the magnetic field component parallel to $\mathbf{k}$ is sufficiently weak, the KHI develops nearly in the flow direction, which corresponds to tailward (anti-sunward) along the magnetopause[16].

In this paper, we show results from a kinetic simulation under realistic magnetopause conditions obtained from the Magnetospheric Multiscale (MMS) spacecraft which feature a super-Alfvénic velocity shear and a weak magnetic shear to satisfy Eq. (1). Past theoretical and numerical studies of the magnetopause suggest that a type of reconnection process, which is induced by the compression of the pre-existing magnetic shear layer (current sheet) by the KHI vortex flow, can give rise to efficient plasma transport along the reconnected field lines[17, 18]. Hereafter we refer to this type of reconnection process[13, 19–22] as vortex-induced reconnection (VIR). The new large-scale 3D simulation demonstrates the turbulent development of VIR within the

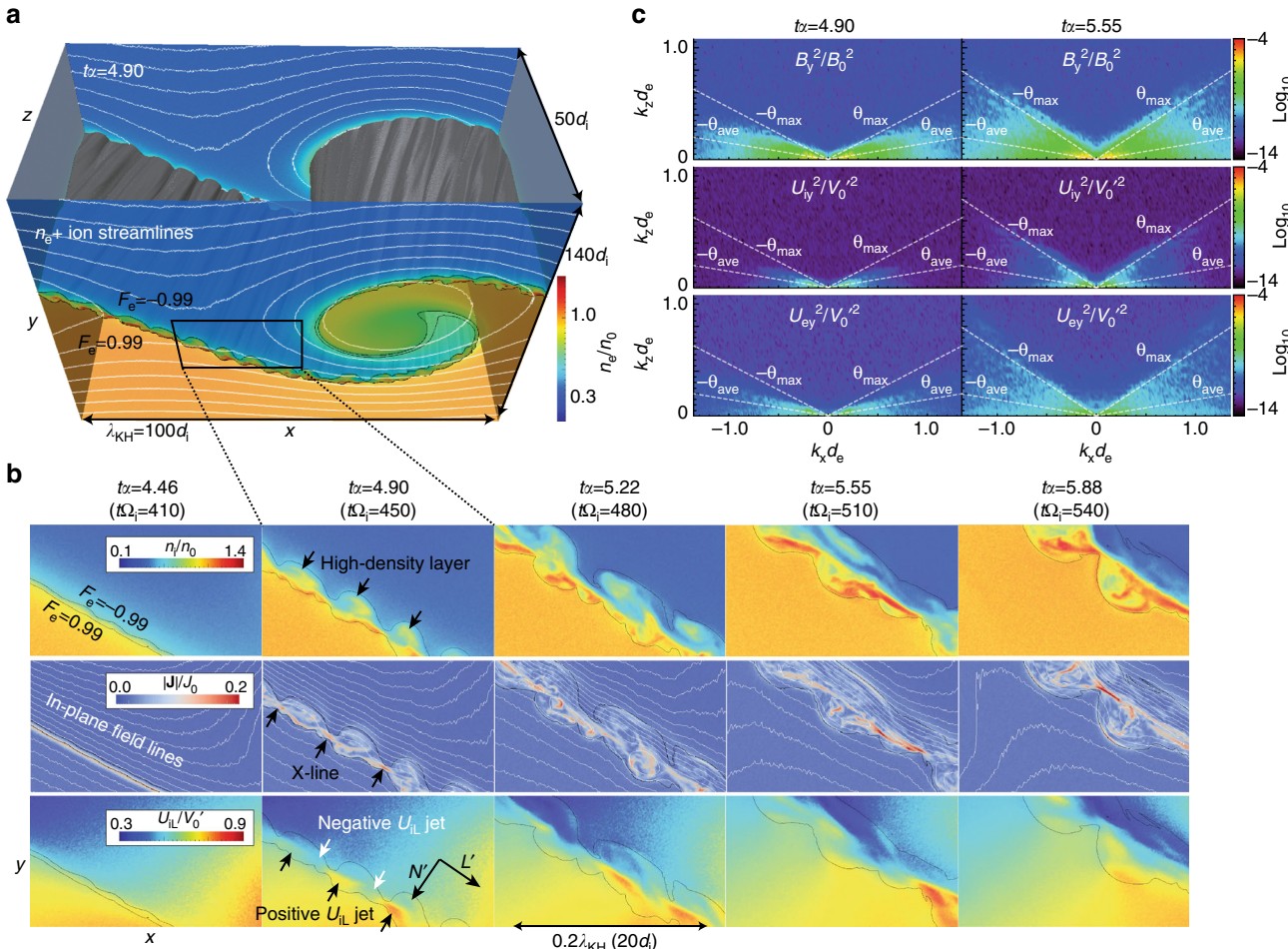

**Fig. 1** Evolution of the ion-scale turbulent reconnection layer produced within the MHD-scale KH vortex. **a** 3D view of mixing surfaces[13, 30] defined as $F_e = (n_{e1} - n_{e2})/(n_{e1} + n_{e2}) = -0.99$ and 0.99 with electron density contours in the x–y planes at $z = 0$ ($=L_z$) in an early non-linear growth phase of the KHI ($t\alpha = 4.90$). The white curves in **a** show ion streamlines projected onto the x–y planes. **b** Zoom-in views of the 2D contours in the x–y plane at $z = 0$ of the ion density $n_i$, the current density $|\mathbf{J}|$ with the in-plane magnetic field lines, and the ion bulk flow component $U_{iL}$ along with the compressed current layer from $t\alpha = 4.57$ to $t\alpha = 5.88$, showing the formation and turbulent decay of ion-scale reconnection signatures. The $L'$ and $N'$ directions marked in **b** show the $L$ and $N$ (parallel and perpendicular to the current layer) directions projected on the x–y plane, respectively. See the Methods section for more details of the LMN coordinates. **c** 2D power spectra ($k_x$, $k_z$) of $B_y$, $U_{iy}$, and $U_{ey}$ at $t\alpha = 4.90$ and 5.50. $\theta_{ave} = \tan^{-1}((B_{in1} + B_{in2})/(B_{out1} + B_{out2}))$ in **c** is the averaged magnetic field angle between the two background regions, showing the expected peak angle satisfying the resonance condition $\mathbf{k} \cdot \mathbf{B} = 0$ of the tearing mode within the boundary region. $\theta_{max}$ in **c** is the maximum oblique angle of the magnetic field in the x–z plane that corresponds to the maximum shear angle of the magnetic field in the x–z plane. The powers of all components are cut off in $\theta > \theta_{max}$, indicating the turbulent development of the 3D tearing mode is the main source of the magnetic field fluctuations as demonstrated in past 3D kinetic simulations of guide-field reconnection[35, 36]

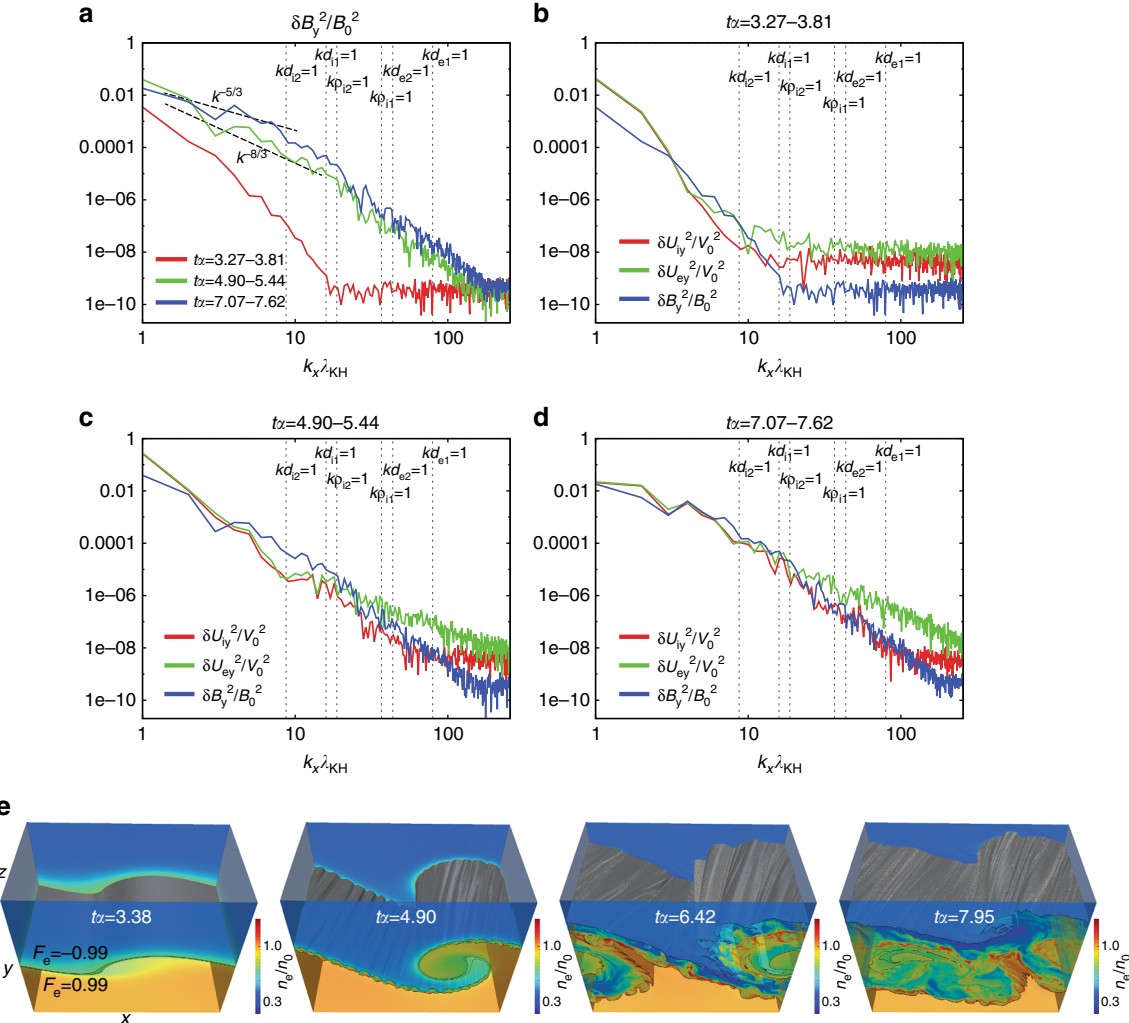

**Fig. 2** Evolution of turbulent spectrum. **a** 1D power spectra ($k_x$) of averaged $B_y$ over $t\alpha = 3.27$–$3.81$ (late linear phase), $t\alpha = 4.90$–$5.44$ (early non-linear phase), and $t\alpha = 7.07$–$7.62$ (late non-linear phase). **b**–**d** 1D power spectra of $U_{iy}$, $U_{ey}$, and $B_y$ averaged over $t\alpha = 3.27$–$3.81$ (**b**) $t\alpha = 4.90$–$5.44$ (**c**), and $t\alpha = 7.07$–$7.62$ (**d**). The blue curves in **b**–**d** are the same ones as the curves shown in **a**. The vertical dashed lines in **a**–**d** indicate the wavelengths for $kd_i = 1$ and $kd_e = 1$ based on $n_0$ ($= n_1$) and $n_2$. Each curve in **a**–**d** is obtained by averaging $U_{iy}$, $U_{ey}$, and $B_y$ of 11 equally spaced time slices, which reduces the short-wavelength particle noise[30]. **e** Time evolution of the 3D view of mixing surfaces from the late linear ($t\alpha = 3.38$) to late non-linear ($t\alpha = 7.95$) growth phase of the KHI

tailward propagating Kelvin–Helmholtz (KH) vortex. This new turbulent phase of VIR is made possible by the larger system size, which permits the more realistic development of the ion-scale features driven by the VIR outflow jets.

## Results

**Evolution of the turbulent vortex layer**. We employed the three-dimensional (3D) fully kinetic simulation code VPIC[23, 24]. The initial parameters are obtained during 0900–1130 UT on 8 September 2015 from the MMS spacecraft, which crossed the magnetopause from the magnetospheric to denser magnetosheath sides, and encountered quasi-periodic KH waves between the two sides[25–29]. The simulation was performed in Cartesian coordinates ($x$, $y$, $z$), in which $y$-direction is perpendicular to the magnetopause, $x$-direction is along the $k$-vector of the fastest growing KH mode, and $z = x \times y$ completes the system. Further details of the initial setup are given in the Methods section.

In this simulation, the equilibrium velocity shear drives a vortex structure as shown in Fig. 1a during the early non-linear growth phase of the KHI. In this phase ($t\alpha \sim 5$), the ion-scale jets of the VIR are formed along the compressed current layer near the

hyperbolic point where the vortex flow is converged (Fig. 1b). Here, $1/\alpha = \lambda_{KH}/V_0$ is the time unit based on the linear growth rate of the KHI[13, 22]. At $t\alpha \sim 4.5$, the electron-scale (less than the ion-inertial length $d_i$) current sheet forms. At $t\alpha \sim 5$, reconnection occurs at multiple sites and this primary reconnection forms ion-scale ($\sim 5d_i$ scale) reconnection outflow jets along the current layer. Since these jets transport low-density plasmas originally located in the upper ($+y$) region, the density within the jets is lower than in the adjacent regions, which forms higher-density layers on the upper (low-density) side. At $t\alpha \sim 5.5$, these ion-scale jet structures rapidly decay and produce a thicker turbulent mixing layer where clear high-density layers can no longer be seen. Figure 1c shows that at $t\alpha = 4.9$, clear peaks of $B_y$, $U_{iy}$, and $U_{ey}$ powers can be seen at $\theta \sim \theta_{ave}$, while at $t\alpha = 5.50$, the powers of all the components are more widely scattered within $\theta < \theta_{max}$. This indicates that the turbulent development of the secondary 3D tearing mode causes the rapid decay of the jet structures as shown in Fig. 1b.

As seen in Fig. 2a, the spectral power in the $y$-component ($B_y$) of the magnetic field is significantly enhanced over the electron ($kd_e \sim 1$) to MHD ($kd_i < 1$) scales in the early non-linear phase (compare the curves for $t\alpha = 3.27$–$3.81$ and $t\alpha = 4.90$–$5.44$). The $B_y$ spectrum for $t\alpha = 4.90$–$5.44$, during which the 3D turbulence

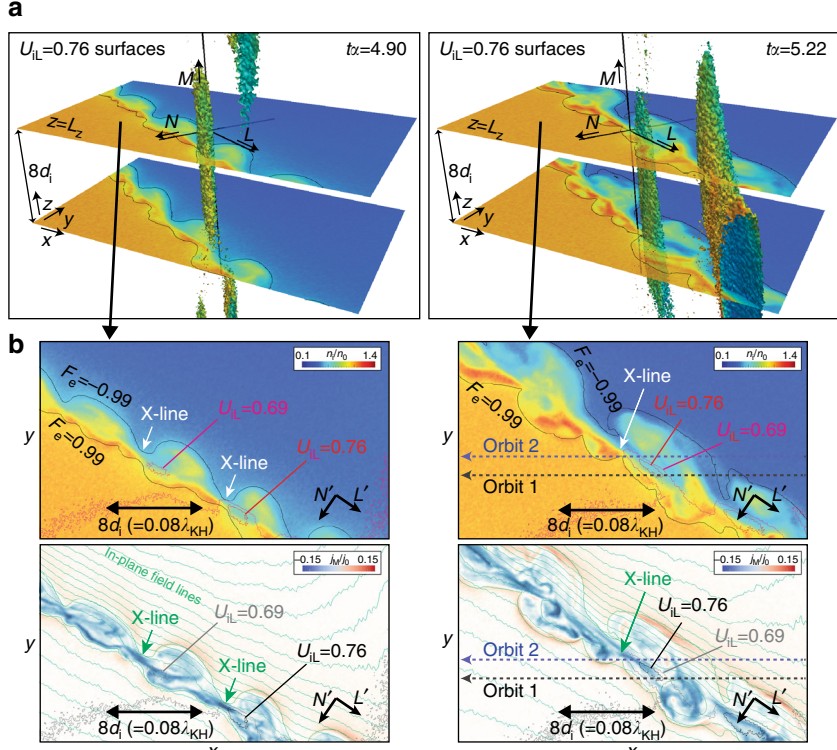

**Fig. 3** Structure of the ion-scale reconnection jets. **a** 3D views at $t\alpha = 4.90$ and $t\alpha = 5.22$ (the times before the turbulent decay of the reconnection layer) of the ion-scale zoomed region (the same region shown in Fig. 1b). The 3D surfaces show the contour surface of $U_{iL} = 0.76 V_0'$, which represents the location and structure of the positive $U_{iL}$ jet in the 3D view. The 2D planes show the ion density in the $x$–$y$ plane at $z = L_z$ ( = 0) and $z = L_z − 8d_i$. The directions of the unit vectors in the $xyz$ system and the local coordinate (LMN) system along the compressed layer are indicated by black arrows. **b** The density and the $M$ component of the current density contours in the $x$–$y$ plane at $z = L_z$ (the same region as the upper slice in **a**). The mixing surfaces and the surfaces of $U_{iL} = 0.76 V_0'$ and $0.69 V_0'$ are shown in both the upper and bottom panels, while the in-plane field lines are shown only in the bottom panels. At both times, the reconnection jet extends in the $L$-direction with a few $d_i$ thickness in the $N$-direction. This jet structure in the $L−N$ plane spreads in the $M$-direction over more than $8d_i$ as shown in **a**. The peak speed and size of the jets in the $L$-direction at $t\alpha = 5.22$ are larger than those at $t\alpha = 4.90$. The density within the jets is lower, leading to the formation of higher-density layers on the magnetospheric ($−N$) side of the jets

of the secondary tearing mode is well developed as seen in Fig. 1b, shows a power law with $\sim k^{-8/3}$ index for the ion ($kd_i \sim 1$) and MHD ($kd_i < 1$) scales and a steeper slope for smaller scales. A turbulent spectrum with a similar slope ($k^{-8/3}$) produced by the secondary 3D tearing mode was seen in past 3D kinetic simulations of guide-field reconnection[30]. In this early non-linear phase, the spectra of the ion and electron flows show significant power enhancements in the ion scale (compare Fig. 2b, c) and clear power laws from the electron to ion scales with a similar index to $B_y$ (Fig. 2c). This supports the above scenario indicated from Fig. 1b, c, in which the structures of the initial ion-scale jets are disturbed by coupling with the turbulent growth of the secondary tearing mode. As seen in Fig. 2e, as the non-linear vortex flow develops, this ion-scale turbulent layer is engulfed into the MHD-scale vortex body, and in the late non-linear phase the whole vortex layer becomes turbulent with well-mixed plasmas. Correspondingly, the spectra in this late non-linear phase feature turbulent power laws with $\sim k^{-5/3}$ index from the ion to MHD scales for all $U_{iy}$, $U_{ey}$, and $B_y$ components (Fig. 2d). This turbulent evolution of the vortex layer leads to a significantly efficient plasma mixing as will be shown later.

**Comparison with the MMS observations for the crossing of reconnection jet.** Figure 3a shows 3D views of the surface of $U_{iL} = 0.76 V_0'$, while Fig. 3b shows 2D views of the surfaces of $U_{iL} = 0.76 V_0'$ and $0.69 V_0'$ in the $x$–$y$ plane at $z = L_z(=0)$. Since the angle of the current layer from the background flow direction in the $x$–$y$

plane (i.e., the angle between $x$ and $L'$) is about 30°, the background flow velocity in the $L$-direction is about $0.5 V_0'$. In addition, since the peak $|B_L|$ around the current layer is about $0.3B_0$ (as will be shown in Fig. 4a), the expected peak speed of the reconnection jet for ions considering the background flow is about $0.5V_0 \pm 0.3 \ V_A \sim (0.5 \pm 0.3)V_0'$. Thus, the isosurface of $U_{iL} = 0.76 \ V_0'$ (and $0.69 \ V_0'$) captures the structure of the ion reconnection jets growing in the positive $L$-direction. At both $t\alpha = 4.90$ and $5.55$, the jets extend in the $L$-direction with a few $d_i$ thickness in the $N$-direction. This 2D-like structure in the $L$–$N$ plane extends in the $M$-direction over more than $8d_i$. The length of the jets in the $L$-direction at $t\alpha = 5.55$ become $1.5$–$2$ times longer than those at $t\alpha = 4.90$.

Figure 4a shows the virtual observation results in which the two probes separated by $1.5d_i$ in the $N$-direction cross the positive $U_{iL}$ jet at $t\alpha = 5.22$ as marked in Fig. 3b. Here, we assume that the jet structures shown at $t\alpha = 5.22$ are not changed and propagate in the background flow direction ($\sim x$-direction) during the crossing by the virtual probes. As seen in Fig. 3b, both probes first observed the density peaks and then observed the $U_{iL}$ peaks with density dips during the crossing of the current layer. The amplitude of $U_{eL}$ at the $U_{iL}$ peaks (vertical lines) is similar to that of $U_{iL}$, but the electron flows tend to have stronger electron-scale ($<d_i$) fluctuations than the ion flows. The density dip for probe-2 which is closer to the X-line is $\sim 1.5$ times deeper and $\sim 2$ times shorter than that for probe-1, indicating that the lower density plasmas within the dips diffuse more strongly in the downstream region of the jets.

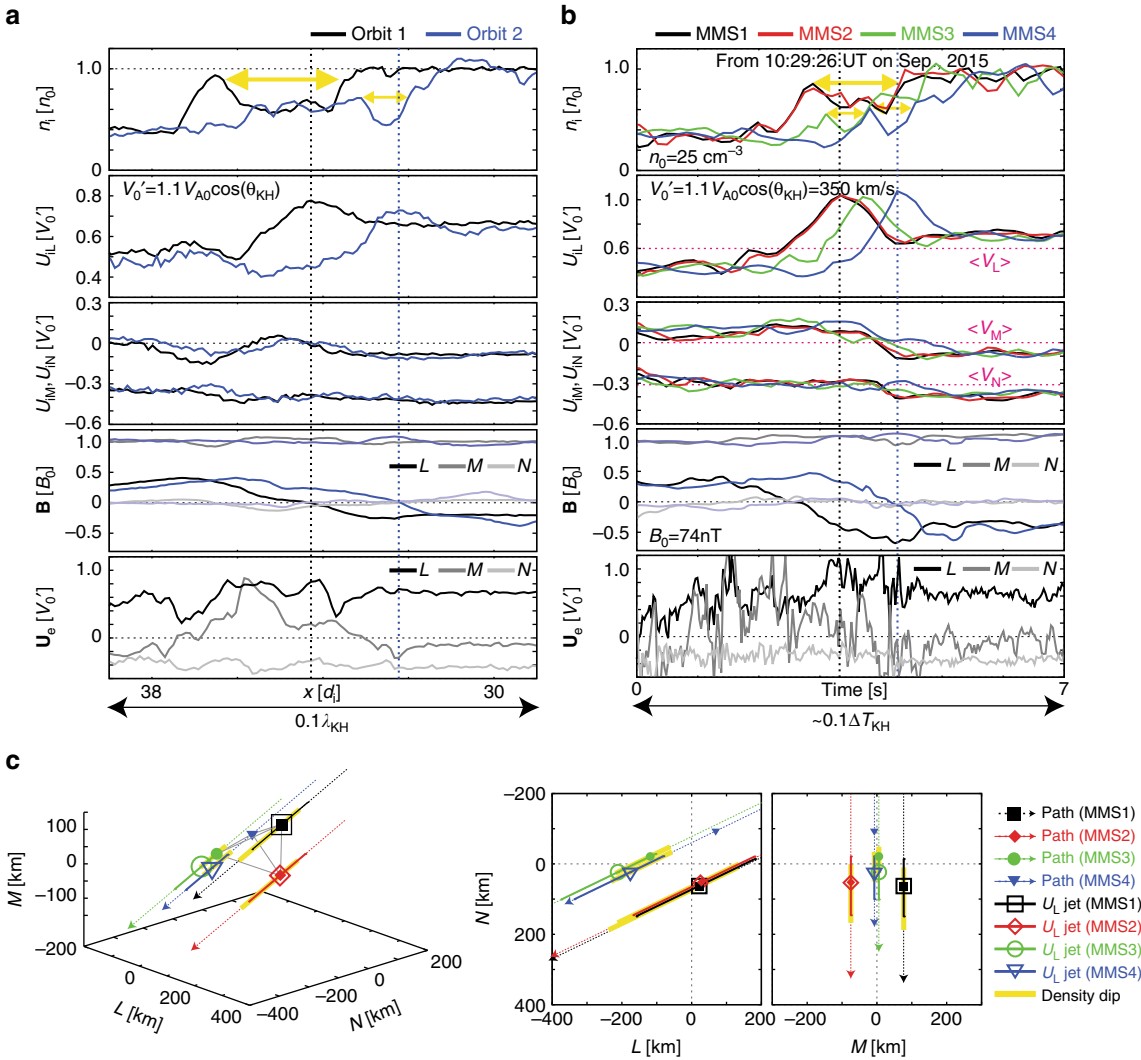

**Fig. 4** Comparison with the in situ observation data from the MMS spacecraft for the crossing of reconnection jets in the KH wave events on 8 September 2015. **a**, **b** Virtual observation for the crossing of a positive $U_{iL}$ jet along the dashed lines shown in Fig. 3b (**a**) and in situ observations by the four MMS spacecraft for a 7 s interval from 10:29:26 UT (**b**) of the ion density, the ion bulk velocity $\mathbf{U_i}$, the magnetic field $\mathbf{B}$, and the electron bulk velocity $\mathbf{U_e}$ velocity components in the LMN system. The data for MMS1 were reported in Fig. 2 in ref. [27]. The values in **a** and **b** are normalized by the initial ($n_0$ and $B_0$) and averaged background ($n_0 = 25$ cm$^{-3}$ and $B_0 = 74$ nT) values in the magnetosheath region, respectively. Vertical lines indicate the location of the $U_{iL}$ peaks for orbits 1 (black) and 2 (blue) in **a** and MMS1 (black) and 4 (blue) in **b**. The yellow arrows indicate the intervals of the density dips. **c** The separation of the four MMS spacecraft on 10:29:26 UT in the LMN system in the 3D view and its projections into the $L-N$ and $M-N$ planes. The closed symbols show relative locations $\mathbf{X_s}$ of the MMS spacecraft. The dashed arrows show the spacecraft paths assuming that the structures move past the MMS tetrahedron at a constant $\langle\mathbf{V}\rangle$ (indicated by the magenta dashed lines in **b**) during this 7 s period. The open symbols and the solid lines show the expected locations $\mathbf{X'_s} = \mathbf{X_s} - \langle\mathbf{V}\rangle(t_s - t_1)$ of the peak and the whole region of the positive $U_{iL}$ jet for each spacecraft relative to the time $t_1$ when MMS1 observed the $U_{iL}$ peak. Similarly, the yellow bars show the expected locations of the density dips

Figure 4b shows the observations from the four MMS spacecraft on 8 September 2015 in the same format as Fig. 4a. Although the data for MMS1 in this event have been reported in ref. [27], here we analyze the data from all four spacecraft. As seen in the simulation, all the spacecraft first observed the density peaks and then the $U_{iL}$ peaks with density dips while crossing the current layer. Notice here that the background ion velocities during this interval are similar among the four spacecraft. Assuming that the jet structures during this interval constantly propagate at the averaged background velocities $\langle\mathbf{V}\rangle$ (magenta lines in Fig. 4b), we can estimate the locations of the $U_{iL}$ jets and their peaks relative to the $U_{iL}$ peak of MMS1 in the LMN system (Fig. 4c). The estimated locations of the jets for the MMS1 and 2 pair and the MMS3 and 4 pair are close in the $L-N$ plane,

respectively. The locations of the two pairs are close in the $N$-direction and separated in the $L$-direction by about 150–200 km, ($\sim 3$–$4 d_i$ for $n_0 = 25$ cm$^{-3}$). The estimated width of the $U_{iL}$ jets in the $N$-direction is about 100–150 km (a few $d_i$). The $M-N$ map indicates that this jet structure in the $L-N$ plane extends in the $M$-direction over more than 200 km ($\sim 4 d_i$). In addition, as shown in Fig. 4b, the density dips of the upstream pair (MMS3 and 4) are $\sim 1.5$ times deeper and $\sim 2$ times shorter than the downstream ones (MMS1 and 2). These features are in quantitative agreement with those seen in the simulation shown in Figs. 3 and 4a especially for the size of the jet (more than $4 d_i$ in the $L$- and $M$-directions with a few $d_i$ width in the $N$-direction) and for the depth and width of the density dips between the two locations separated by about $4 d_i$ in the $L$-direction.

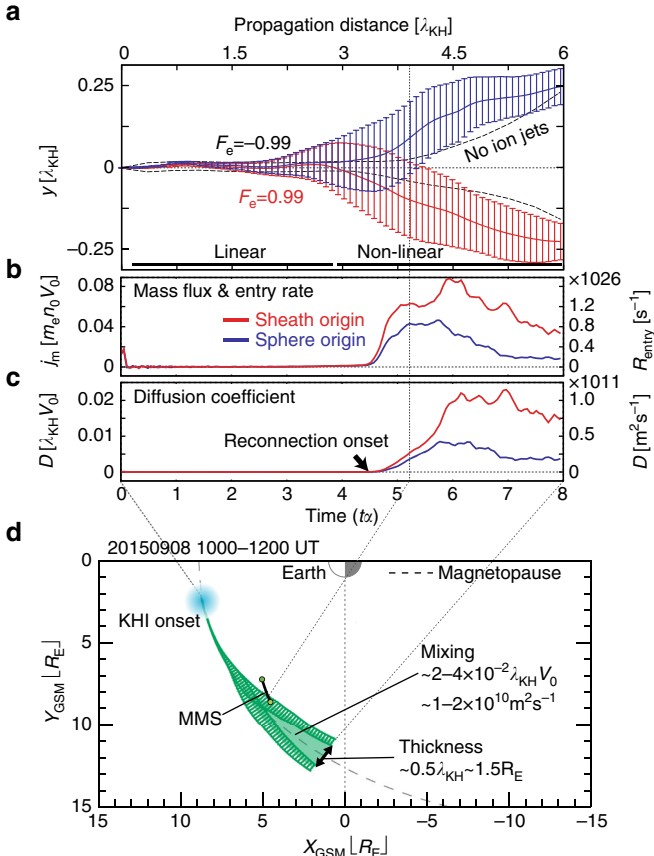

**Fig. 5** Time evolution of the plasma mixing. **a–c** Time evolution of **a** the locations in y of the mixing surfaces (defined by $|F_e| = 0.99$ as defined in ref. [13]) averaged along the x and z-direction with standard deviations, **b** the mass flux $j_m = (d\Sigma m/dt)/(L_x L_z)$ computed from the time variation of the integrated mass entering the mixing region across the mixing surfaces, and the corresponding entry rate per one wavelength of the KH vortex $R_{entry} = j_m m_e \lambda_{KH}^2$, of the particles (electrons) originally located in the magnetosheath (red) and the magnetosphere (blue), and **c** the diffusion coefficient D estimated from the Fick's law $j_m/D = d(m_e n_e)/dx \sim m_e n_0/L_{mix}$ where $L_{mix}$ is the averaged thickness in y of the mixing region. The dashed line in **a** is of the smaller vortex case shown in ref. [13]. The entry rate in **b** is obtained by assuming that the thickness of the KHI-active region in the $Z_{GSM}$ direction is $\sim\lambda_{KH}$. **d** MMS trajectory in the $X_{GSM}$–$Y_{GSM}$ plane during 1000–1200 UT on 8 September 2015 with the model magnetopause defined in ref. [51]. The green regions in **d** are the locations of the mixing surfaces shown in **a** projected onto the modeled magnetopause. The entry rate in the $s^{-1}$ unit in **b**, the diffusion coefficient in the $m^2 s^{-1}$ unit in **c** and the special scale for the projection in **d** are obtained by using the typical parameters during the event; $n_0 = 25$ cm$^{-3}$, $V_0 = 355$ km s$^{-1}$, and $\lambda_{KH} = V_{KH}\Delta t_{KH} \sim 15{,}000$ km

**Evolution of plasma mixing within the turbulent vortex layer.** These quantitative consistencies between the simulation and observation strongly suggest that the formation of the ion-scale VIR jets and the associated high-density layers on the low-density (magnetospheric) side most likely occurred in this MMS event. Since these density structures tend to rapidly decay within $\Delta t\alpha < 1$ in the simulation (Fig. 1b), the consistencies naturally suggest that the KHI at the MMS location was in the early non-linear phase ($t\alpha \sim 5$) as shown in the second panel in Fig. 2e. Based on this evidence, we can estimate the mass transfer across the magnetopause for this event. As seen in Fig. 5a, the mixing region rapidly expands after the onset of the VIR. Assuming that the KH vortex propagates along the magnetopause at the phase speed of

the KHI and the background shearing flow is constant (i.e., not globally changed), the simulation time can be converted to the propagation distance from the onset location[13]. This propagation distance predicts the formation of the mixing layer with thickness $1.5R_E$ at $0 < X_{GSM} < 5R_E$ (Fig. 5d). The mass flux and particle entry rate into the mixing region and the corresponding diffusion coefficient start to increase at around the MMS location (Fig. 5b, c), due to the evolution of the VIR. The entry rate of the solar wind (magentosheath) particles reaches $R_{entry} \sim N_{KH} \times 1.0 \times 10^{26}$ s$^{-1}$, where $N_{KH}$ is the number of KH vortices simultaneously generated along the magnetopause. This value is comparable or larger than the rate resulting from reconnection for a large magnetic shear ($R_{entry} \sim 10^{24}$–$10^{27}$ s$^{-1}$), which was estimated from in situ observations[31, 32]. The diffusion coefficient reaches $D \sim 0.7$–$1.0 \times 10^{11}$ m$^2$ s$^{-1}$ for the magnetosheath particles and $D \sim 0.2$–$0.5 \times 10^{11}$ m$^2$ s$^{-1}$ for the magnetospheric particles. The theoretically required diffusion coefficient[33] for populating the mixing layer, which is frequently observed along the low-latitude magnetopause is $D \sim 10^9$ m$^2$ s$^{-1}$. In addition, past simulation and observational studies of the KHI excited at the magnetopause[8, 10–13, 34] predicted similar values in the range about $D \sim 10^9$–$10^{10}$ m$^2$ s$^{-1}$. However, the values from the present simulation ($D \sim 10^{11}$ m$^2$ s$^{-1}$) are almost one order higher than these past predictions for the KHI (Fig. 5c). As illustrated in Fig. 5d, this may contribute to the formation of a thicker mixing layer along the more distant magnetopause than previous predictions[8, 13, 16].

## Discussion

The previous smaller-scale kinetic simulations of VIR[22], which also showed nearly one order smaller D ($\sim 10^{10}$ m$^2$ s$^{-1}$ as shown in Supplementary Fig. 1b), predicted that the typical distance between well-developed primary X-lines (i.e., the typical primary flux rope size) scales with $\lambda_{KH}$ and is of the order of $\sim 0.1\lambda_{KH}$ (cf., Supplementary Fig. 1). This is consistent with the distance ($\sim 8d_i = 0.08\lambda_{KH}$) of the present larger-scale simulation (Fig. 3b). Based on this scaling, assuming that ion-scale reconnection jets are strongly suppressed when the available room on each side of X-line is less than a few $d_i$, the condition for the strong growth of ion-scale jets can roughly be predicted as

$$\lambda_{KH} > \sim 50d_i. \qquad (2)$$

This prediction naturally explains why the ion-scale jets could not be seen in the previous smaller-scale simulations[13, 22] ($\lambda_{KH} = 15$–$30d_i$) as shown in Supplementary Fig. 1b, but can be observed in the present simulation ($\lambda_{KH} = 100d_i$) as well as the MMS event analyzed in this paper ($\lambda_{KH} \sim 300d_i$[25]).

In addition, the past 3D kinetic simulations of guide-field reconnection[30, 35] suggested that the structure of an ion-scale primary reconnection layer is strongly disturbed by the turbulent development of the secondary 3D tearing mode within $\Delta t \sim 30$–$50\ \Omega_i^{-1}$ where $\Omega_i^{-1}$ is the ion gyrofrequency based on $B_0$. This implies that the necessary condition for the development of primary flux rope structures from VIR decay during the early non-linear phase (i.e., the time between $t\alpha \sim 4.5$–6) is roughly

$$\alpha^{-1} = \frac{\lambda_{KH}}{V_0} > \sim 50\Omega_i^{-1}. \qquad (3)$$

This estimate naturally explains why the primary jet structures (and the related primary flux rope structures) in the present simulation ($\alpha^{-1} \sim 90\ \Omega_i^{-1}$) decay within the early non-linear phase, but the primary ropes in the previous smaller-scale simulations[13, 22] ($\alpha^{-1} \sim 11$–$21\ \Omega_i^{-1}$) can survive until the late non-linear phase (cf., Supplementary Fig. 1b). Since $\alpha^{-1} \sim 300\ \Omega_i^{-1}$ in this particular MMS event, the conditions satisfy the above decay constraint for the primary structures. Note that the

initial source of both the primary jet and secondary turbulence is electron-scale tearing modes excited within thin electron-scale current layers[21, 22, 30, 35, 36]. Past kinetic studies suggested that electron-scale tearing modes (flux ropes) rapidly merge into ion-scale flux ropes within a short time ($\Delta t <$ a few $\Omega_i^{-1}$)[35, 37]. Thus the formation and decay processes of the ion-scale jets are not significantly affected by the initial electron-scale modes, as shown in Supplementary Fig. 2c.

In summary, as predicted in Eqs. (2) and (3), the formation and decay of ion-scale VIR jet structures can occur in the early non-linear growth phase of the KHI when the spatial and growth time scales of the KHI are much larger than the ion scales ($\lambda_{KH} \gg d_i$, $\alpha^{-1} \gg \Omega_i^{-1}$). These conditions are easily satisfied for typical encounters of KH waves in the Earth's magnetopause including the MMS event analyzed in this paper. While there are a number of simulation studies of the KHI using single and multi-fluid[8, 17–19, 38, 39] and hybrid kinetic-ion/electron-fluid[12, 40] simulations, fully kinetic 3D simulations are required to accurately describe the physics of VIR[21], since this approach permits a realistic spectrum of secondary tearing modes to develop[22] and modify the plasma transport[30]. However, resolving the broad range of scales in fully kinetic simulations is very challenging, even for the newest generation of petascale computers. The fully kinetic 3D results presented in this letter represent the first such example, allowing us to quantitatively investigate the ion-scale VIR physics for the first time. The resulting mass transfer rate is nearly one order of magnitude larger than previous small-scale 3D kinetic simulations[13, 22] (cf., Supplementary Fig. 2b). In addition, the rapid variations of plasma moments produced by the ion-scale jets seen in Fig. 4b can only be resolved using the new high-resolution MMS measurements[41]. Taken together, the present simulation and observational results suggest that the mass transfer from the VIR may be significantly stronger than previously thought. This finding is crucial for understanding how the solar wind enters the Earth's magnetosphere, but may also be important for other planetary magnetospheres, or the thin boundary layer at the outer heliosphere. Finally, these simulations may be important for understanding the general problem of dissipation and transport in turbulent kinetic plasmas[42], and in the future it will be useful to compare with other turbulent processes with similar power law indices ($\sim k^{-8/3}$ to $k^{-5/3}$), such as magnetic reconnection in the magnetotail[43–45] and turbulence in the magnetosheath[46–48] and solar wind[49, 50].

## Methods

**Simulation settings.** The simulation was performed on the Titan machine at the Oak Ridge Leadership Computing Facility, using the high-performance 3D fully kinetic particle-in-cell code VPIC[23, 24], which solves the relativistic Vlasov–Maxwell system of equations. The initial density, magnetic field, and ion bulk velocities between the two regions across the boundary are set up by referring to the values obtained from the MMS observations in the magnetosphere and the magnetosheath before and after the interval 1010–1120 UT on 8 September 2015 during which the quasi-periodic KH waves were observed[25–29]. Denoting the higher and lower density sides as 1 and 2, we first chose the density ($n_1$, $n_2$), the magnetic field ($\mathbf{B}_1$, $\mathbf{B}_2$), and the bulk velocities ($\mathbf{U}_1$, $\mathbf{U}_2$), in coordinates ($x'$, $y'$, $z'$) where $x'$ is the direction of the shearing flow whose amplitude is $V_0$, $y'$ is the boundary normal using the magnetopause model[51], and $z'$ is obtained from $\mathbf{e}'_x \times \mathbf{e}'_y$. Then, the values were converted to coordinates ($x$, $y = y'$, $z$) along the $k$-vector of the fastest growing mode of the KHI[52], in which the $x$-axis is rotated by $-8.3$ degrees from the $x'$-direction in the $x$–$z$ plane. The rotation angle is obtained from Eq. (1) by substituting the above values in regions 1 and 2. The obtained set of values used in the simulation are $n_2/n_1 = 0.3$, $(B_{x1}, B_{y1}, B_{z1}) = (-0.1B_0, 0, B_0)$, $(B_{x2}, B_{y2}, B_{z2}) = (0.2B_0, 0, B_0)$, $(U_{x1}, U_{y1}, U_{z1}) = (V_0\cos(8.3°), 0, V_0\sin(8.3°))$, and $(U_{x2}, U_{y2}, U_{z2}) = (0, 0, 0)$, where $n_1 = n_0 = 25$ cm$^{-3}$, $B_0 = 74$ nT, and $|V_0| = 355$ km s$^{-1}$ = 1.1 $V_A$ based on $n_0$ and $B_0$. The $y$-component of the magnetic field and velocities were neglected, since these are smaller than the $x$ and $z$ components. The ion density, the magnetic field, and the ion and electron bulk velocities are prescribed by connecting the above values using the $\tanh(y/L)$ function[13], where $L = 6.67d_i$ is the initial thickness of the shear layer. The electron temperature is set to be uniform, while the ion temperature is set to satisfy the pressure balance, where the ion-to-

electron temperature ratios in region 1 and 2 are set to be $T_{i1}/T_{e0} = 3.0$ and $T_{i2}/T_{e0} = 11.53$, respectively. The total plasma $\beta$ in region 1 and 2 are 0.50 and 0.53, respectively. The additional electron and ion flows are added to satisfy the shifted Harris type current sheet of the $B_x$ component. The electric field is set to satisfy $\mathbf{E} = -\mathbf{U}_e \times \mathbf{B}$, and the electron density is set to be slightly higher than the ion density in the shear layer to satisfy the Gauss's law[53]. The ion-to-electron mass ratio $m_i/m_e = 25$, and the ratio between the electron plasma frequency and the gyrofrequency based on $n_0$ and $B_0$ $\omega_{pe}/\Omega_e = 1.0$. The system size is $L_x \times L_y \times L_z = 100d_i \times 150d_i \times 50d_i = 2048 \times 3072 \times 1024$ cells with a total of $1.3 \times 10^{12}$ superparticles. The system length $L_x = 15L$ corresponds to the wavelength of the fastest growing KH mode[6], which is a few times smaller than the observed wavelength. The system is periodic in the $x$- and $z$-directions, and the $y$-boundaries are modeled as perfect conductors for the fields and reflecting for the particles.

**MMS observations and local coordinate systems.** The in situ observation data shown in Fig. 4b were obtained from the MMS spacecraft during the same interval (around 10:26:30 UT on 8 September 2015) as shown in Fig. 2 in ref. [27]. Although ref. [27] only showed the data from MMS1, this study required a multi-spacecraft analysis of the data from all four MMS spacecraft. The local coordinates (LMN) in Fig. 4b are also obtained in the same manner as Fig. 2 in ref. [27], where the normal direction $\mathbf{N}$ (= (0.95, 0.08, −0.31) in GSM) is obtained by the timing analysis of the magnetic field data, the $\mathbf{M}$ (=(0.21, 0.58, 0.79) in GSM) direction is defined by the cross product between $\mathbf{N}$ and the maximum variance direction of the magnetic field[54] across the boundary and $\mathbf{L} = \mathbf{M} \times \mathbf{N}$ (=(−0.25, 0.81, −0.54) in GSM) completes the system. See ref. [27] for further details for the MMS observation methods in this event.

For the simulation, the local coordinates (in the zoomed-in-views of the VIR regions shown in Figs. 1b and 3 and the virtual observation plot in Fig. 4a) are obtained in the same procedure as the observation, except that $\mathbf{N}$ is obtained as the cross product normal[54] $\mathbf{N} = (\mathbf{B}_a \times \mathbf{B}_b)/|\mathbf{B}_a \times \mathbf{B}_b|$ where $\mathbf{B}_a$ and $\mathbf{B}_b$ are the magnetic field shortly after and before the boundary crossing, respectively. The $L'$- and $N'$-directions marked in Figs. 1b and 3b show the $L$- and $N$-directions, respectively, projected in the $x$–$y$ plane.

**Data availability**. The simulation data that support the findings of this study are available via the Oak Ridge Leadership Computing Facility (OLCF) repository (https://doi.ccs.ornl.gov/ui/doi/46). The observational portion of this research uses the data from the MMS spacecraft, which are publically available via NASA resources and the Science Data Center at CU/LASP (https://lasp.colorado.edu/mms/sdc/public/).

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

## Acknowledgements

We gratefully acknowledge support from the Austrian Research Fund (FWF): I2016-N20. The simulation part of this research used resources of the Oak Ridge Leadership Computing Facility, which is a DOE Office of Science User Facility supported under Contract DE-AC05-00OR22725. The simulation data were also analyzed with resources at the Space Research Institute of Austrian Academy of Sciences. Contributions from W. D. were supported by NASA grant NNH13AV26I. Contributions from S. E. were supported by NASA MMS-Phase E support to CU/LASP and NASA grants NNX08AO84G and NNH13AV26I. We thank T. Nagai, Y. Narita, S. Zenitani K. Genestreti, and M.H. Spencer for their fruitful discussions.

## Author contributions

T.K.M.N. carried out the kinetic simulation and the multipoint data analysis, and wrote the paper. T.K.M.N., H.H. and W.D. contributed to the analysis of the simulation data. H.H., S.E. W.L. and R.N. contributed to handling the observation data. S.E. contributed to set up the simulation. W.D. supported the simulation performed on Titan, and contributed to writing and editing the manuscript. All of the authors discussed the results and commented on the paper.

## Additional information

**Competing interests:** The authors declare no competing financial interests.

