## [Peer Review File · Nature Communications]

Reviewers' comments:

Reviewer #1 (Remarks to the Author):

Nature Comm. Nakamura, TKM et al., Turbulent mass transfer caused by vortex-induced reconnection in collisionless magnetospheric plasmas

The subject of this paper is very important for magnetospheric physics. The results obtained by complex numerical simulations are new, and interesting. The process was conjectured in the late nineties, dealt by first studies at the beginning of the present century, and acquired full status in the last decade. The paper adds even more substance on the importance of KHI vortices for mass transfer into the magnetosphere (something considered heretic not more than twenty years ago). The present work helps the analysis of the very high resolution MMS data (one of the most important present day space missions) another point of merit. The authors are professional space physicists, with experience in the matter, which work with well known space institutions. The paper readability is perhaps not at its top, but not wishing to cause further delays, I recommend undeservedly publication as it is.

Reviewer #2 (Remarks to the Author):

Review Report on "Turbulent mass transfer caused by vortex-induced reconnection in collisionless magnetospheric plasma"

Overview of the paper:

Paper compares virtual spacecraft signatures during large-scale (100 d_i~15000 km) kinetic simulations of the Kelvin-Helmholtz Instability between MMS spacecraft observations and quantifies the associated plasma transport. For this case, the magnetic field components along the shear flow were initially anti-parallel in the magnetosheath and magnetospheric-sides, respectively. The plasma mixing and transport arises because of the decay of the ion-scale jets that were formed by magnetic reconnection. Main conclusion is that new transport estimates are nearly one order higher than previous mass transport calculations in relation to reconnection and diffusion in KH vortices.

General Comments:

The paper addresses a fundamentally important physical mechanism responsible for plasma transport between magnetosheath and magnetospheric plasma. I think the simulations re-produce particularly well the corresponding MMS observations related to ion dynamics, of this well studied KH-event. I think the paper is therefore well suitable for Nature Communications due to its universal nature: these results are applicable and highly relevant to many magnetized plasma systems that exhibit velocity shear.

However, I strongly recommend following revisions and clarifications before publication is granted. In particular, related to existing literature, some references need to be re-organized, and some further references need to be cited. Especially discuss more clearly how transport is quantified in previous studies and how it is done here (as this may contribute to some of the differences). In order to address the effect of electron to ion mass ratio to the mass transport calculations, I would recommend to have a supplementary information section, discussing simulation results with higher mass ratio and higher spatial resolution to evaluate the robustness of the presented results. Alternatively, if higher mass ratio is difficult to consider due to computational resources required, I recommend discussing the results when smaller mass ratio is used to see how results change when compared to higher mass ratios. Please see detailed comments below.

Specific Comments:

Title: If turbulence is not verified by the existence of some turbulent power law in k-space in simulations, the title should probably be changed to "Mass transfer caused by vortex-induced reconnection in collisionless magnetospheric plasmas".

In principle, it would be easy and quite interesting to have a cut through system at various stages of the development and compute the magnetic field and electron and ion velocity field power spectrum in k-space to see how the power law varies to justify the word "turbulent" in the original title.

1. Line 37: references 5-8 should be added after word (KHI) or re-organized, as not all of these references (5-8) address or quantify the plasma transport related to KHI but discuss various other aspects of the KHI. For example, add references 5,6,8 after word "(KHI)", then references 25, 7 and at least few for each category (i-iii) of the following missing ones after the word "considerable transport":

i) Kinetic scale mixing in association with KHI:

Fujimoto and Terasawa, 1991, 1994, 1995

Thomas and Winske 1993

Huba 1996

Matsumoto and Hoshino 2006b

Cowee et al. 2009

Delamere et al, 2011

Nakamura and Daughton, 2014

ii) Blobby transport in association with 2-D and 3-D reconnection in KH vortices:

Ref. 25 and Nykyri and Otto, 2004 (2-D MHD and Hall-MHD)

Ma et al. 2014a,b (3-D KHI and reconnection, with either KHI or reconnection as primary process)

iii) Transport via coupling through KHI and kinetic scale waves, e.g Kinetic Alfvén waves:

Chaston et al., 2007;

Johnson, J. R., C. Z. Cheng, and P. Song (1997).

2.

Lines 31-46: The discussion related to KH driven plasma transport is rather short and does not differentiate well between

i). Macroscopic "blobby" transport (Type B) due to KH driven reconnection (when magnetic field components are initially parallel across shear flow layer, and where KHI as a primary mechanism generates anti-parallel components leading to reconnection and between

ii). The Type A (This present MMS case and Nykyri et al. 2006, angeo Cluster case) where magnetic fields were initially anti-parallel between magnetosheath and magnetospheric field, where the twisting of the boundary by KHI enables reconnection, because flow becomes sub-Alfvénic.

I believe this Type A aspect was first clearly addressed in the following simulation paper: Chen, Q., A. Otto, and L. C. Lee (1997). Tearing instability, Kelvin-Helmholtz instability, and magnetic reconnection. *J. Geophys. Res.* 102, 151-161 and should be referenced.

3.

Line 69: How is "Turbulent development" determined. Was there a specific turbulent power law that was observed in the simulations? What is the detailed physical mechanism for jet decay, e.g did the jets excite some secondary instability or kinetic wave modes allowing for the secondary mixing to happen?

4. Lines 71-81 and Figure 1 & 4: I understand that the mass flux across the mixing surfaces j_m is computed using $j_m \sim m_e n_0 V_0$, where V_0 is the velocity shear of 355 km/s. And the large

value for the j_m is obtained because the ion jets grow mainly along L-direction, with $U_{iL}=0.76 V_0$. However, the U_{iN} would be much smaller and would therefore give more modest transport rates if transport from magnetosheath into magnetosphere along N-direction was considered. Based on the Figure 1b, it looks like the width of the transport layer along N direction is only about $(3-4)d_i$, so while efficient mixing along L is occurring it is initially localized to quite narrow area. In Figure 4 it is clear after $t_a=7.95$ that a larger area gets affected with a width (along N) comparable to the amplitude of the fastest growing KH mode. However, I think the $U_{iL}=0.76 V_0$ value used, possibly overestimates the transport and perhaps a better way to estimate the maximum transport at $t_a=7.95$ from magnetosheath to the magnetosphere would be to use following "entry"-velocity $V=(n_e*m_e*Area)/(rho_{sheath}*t*\lambda)$, and the diffusion coefficient would be then given by multiplying this velocity by the thickness of the velocity shear layer or the amplitude of the large-scale KH mode. Looking at Figure 4 at $t_a=6.42$, it is obvious that if this simulation was generated for the conditions with initially parallel-B-fields across shear flow layer (Type B reconnection in KH vortices), the reconnection would occur in high plasma density current layer resulting in "blobby" transport of two larger scale magnetic islands giving roughly same order of value of $V=(n_e*m_e*Area)/(rho_{sheath}*t*\lambda)$ as estimated here by eye for present results.

This issue should be clarified in the revised version.

5. Lines 127-130: Discuss here or in the beginning (see comment 1.) more thoroughly the literature related to plasma transport calculations: what methods and plasma approximations were used and what rates were obtained using MHD, Hall-MHD, hybrid, full particle etc. List the relevant missing references: e.g, Nykyri and Otto, 2004, Angeo (Hall-MHD), Cowee et al, 2009, 2010 (hybrid), etc.

6. Lines: 157-158: Please denote, the value of the electron and ion plasma beta in the magnetosheath and magnetosphere, as this can have an effect on what kind of kinetic plasma wave modes could be excited into the system.

7. Line 167: electron to ion mass ratio of 25 is used. Address, how would a more realistic mass ratio affect the results? For example, either by making one simulation with higher or lower mass ratio than for the presented case and discussing the results. I presume that for realistic mass ratio the electrons would stay magnetized longer, and in order to resolve the electron-scale kinetic wave-modes that could contribute to electron de-magnetization, a very high resolution run would be required to resolve the electron-scale wave length waves. Here the numerical resolution of 7 km is appropriate for mass ratio of 25, but probably insufficient if addressing a mass ratio of 1836. Include in methods some discussion related to electron de-magnetization physics, numerical resolution and mass ratios. What terms/physics mostly contribute to the breaking of the frozen in condition for electrons in these simulations (e.g plasma wave modes producing anomalous resistivity, electron inertia etc., off-diagonal components of electron pressure tensor). If you think the details of electron reconnection physics are not important for ion-scale jet generation and subsequent transport, it would be good to explicitly state and discuss this briefly.

8. Magnetic field vectors are missing in Figure 1b (current density plot), as adding these would make this plot difficult to read, perhaps make one of the figures larger (or put into supplementary information section) and draw the b-field vectors into it. This makes it easier for the reader to understand where the largest shear occurs at the initial system and how the shear evolves.

9. Line 154-156: This method to estimate the rotation angle to the direction of the k-vector of the fastest growing mode has been used in 3-D KH simulation studies which was created after Cluster KHI event (Nykyri et al., 2006, angeo), with similar magnetic field geometry across shear flow plane as for the presented MMS event, and these should be referenced here:

Adamson E., Nykyri K., Otto A., The Kelvin-Helmholtz Instability Under Parker-Spiral Interplanetary Magnetic ... 218-230, doi:10.1016/j.asr.2015.09.013., 2016.

Reviewer #3 (Remarks to the Author):

Report on "Turbulent mass transfer caused by vortex-induced reconnection in collisionless magnetospheric plasmas" by Nakamura et al.

The authors performed one new larger-scale kinetic simulation to investigate the evolution of Kelvin-Helmholtz instability (KHI) and presented ion-scale jets from the vortex-induced reconnection (VIR) rapidly decay through self-generated turbulent reconnection. The authors also compared with in-situ observations by MMS mission. In addition, the authors estimated mass transfer rate nearly one-order higher than previous expectations for the KHI. This manuscript gives some new results that can help to understand how solar wind enters the planetary magnetosphere. However, except the large-scale simulation gives some features of ion-scale jet and the higher mass transfer rate, the author did not present novel findings to meet the standards for publication in Nature Communication. Thus, I think the manuscript may not be suitable for publication in Nature Communication. Below I will give some main comments.

1. The authors claimed that mass transfer rate is nearly one-order higher than previous simulations. Compared this simulation with the authors' previous simulations [Ref 14], only difference is the box size. Why mass transfer rate increases one order with the increase of the box size? That's so confuse! The authors should give some physical explanations.
2. The authors didn't propose any new mechanisms for the mass transfer from solar wind into the magnetosphere. Their manuscript still investigates Kelvin-Helmholtz instability which is well-known to play curial role on the mass transport across the magnetopause.
3. The author emphasized the important role of turbulent mass transfer caused by vortex-induced reconnection. However, they didn't show the evidence of self-generated turbulent reconnection, and didn't explain what's turbulent mass transfer and what's the difference with previous mass transfer mechanism.

RESPONSE TO REFEREES

We would like to thank the three reviewers for their careful reading of our paper. We have taken these comments into account in the revised manuscript. Below are our responses to each comment.

Reviewer #1:

The subject of this paper is very important for magnetospheric physics. The results obtained by complex numerical simulations are new, and interesting. The process was conjectured in the late nineties, dealt by first studies at the beginning of the present century, and acquired full status in the last decade. The paper adds even more substance on the importance of KHI vortices for mass transfer into the magnetosphere (something considered heretic not more than twenty years ago). The present work helps the analysis of the very high resolution MMS data (one of the most important present day space missions) another point of merit. The authors are professional space physicists, with experience in the matter, which work with well known space institutions. The paper readability is perhaps not at its top, but not wishing to cause further delays, I recommend undeservedly publication as it is.

We sincerely appreciate the reviewer's positive and complementary comments. In the revised manuscript, we have attempted to improve the readability.

Reviewer #2:

Overview of the paper:

Paper compares virtual spacecraft signatures during large-scale (100 d_i~15000 km) kinetic simulations of the Kelvin-Helmholtz Instability between MMS spacecraft observations and quantifies the associated plasma transport. For this case, the magnetic field components along the shear flow were initially anti-parallel in the magnetosheath and magnetospheric-sides, respectively. The plasma mixing and transport arises because of the decay of the ion-scale jets that were formed by magnetic reconnection. Main conclusion is that new transport estimates are nearly one order higher than previous mass transport calculations in relation to reconnection and diffusion in KH vortices.

We sincerely appreciate the reviewer's understanding and complementary comments.

General Comments:

The paper addresses a fundamentally important physical mechanism responsible for plasma transport between magnetosheath and magnetospheric plasma. I think the simulations re-produce particularly well the corresponding MMS observations related to ion dynamics, of this well studied KH-event. I think the paper is therefore well suitable for Nature Communications due to its universal nature: these results are applicable and highly relevant to many magnetized plasma systems that exhibit velocity shear.

However, I strongly recommend following revisions and clarifications before publication is granted. In particular, related to existing literature, some references need to be re-organized, and some further

references need to be cited. Especially discuss more clearly how transport is quantified in previous studies and how it is done here (as this may contribute to some of the differences).

In order to address the effect of electron to ion mass ratio to the mass transport calculations, I would recommend to have a supplementary information section, discussing simulation results with higher mass ratio and higher spatial resolution to evaluate the robustness of the presented results.

Alternatively, if higher mass ratio is difficult to consider due to computational resources required, I recommend discussing the results when smaller mass ratio is used to see how results change when compared to higher mass ratios. Please see detailed comments below.

We appreciate the reviewer's useful suggestions. We basically agree with the suggestions. We newly referred to many of the suggested papers and re-organized the references to highlight the difference of this paper from the past studies. In addition, we newly created the Supplementary information section to discuss the mass ratio effects as well as the dependences on the spatial size.

Specific Comments:

Title: If turbulence is not verified by the existence of some turbulent power law in k-space in simulations, the title should probably be changed to "Mass transfer caused by vortex-induced reconnection in collisionless magnetospheric plasmas".

In principle, it would be easy and quite interesting to have a cut through system at various stages of the development and compute the magnetic field and electron and ion velocity field power spectrum in k-space to see how the power law varies to justify the word "turbulent" in the original title.

Taking into account the reviewer's comment, we newly added a figure (Fig.2) showing the power spectra of the magnetic field and the ion and electron flows. The spectra show that the magnetic field power is enhanced from the electron to MHD scales with a clear power law in the early non-linear phase. This indicates the turbulent development of the secondary 3D tearing mode. The ion and electron flow powers are enhanced mainly at the ion-scale and show similar power laws to the magnetic field from the electron to ion scales in this early non-linear phase. This indicates that the ion-scale jet structures are disturbed by the turbulence of the secondary tearing mode. In the late non-linear phase, the flow and field powers show clear power laws with a $-5/3$ index from the ion to MHD scales, as the ion-scale turbulent layer is engulfed into the vortex body and the whole vortex layer becomes turbulent. Thus, the spectra shown in the new Fig. 2 supports the main conclusion of this paper on the formation and turbulent decay of the ion-scale reconnection jet structures and the subsequent efficient plasma transport. Based on this, we keep "turbulent" in the title. We have added a new paragraph on this topic in lines 73-89.

1. Line 37: references 5-8 should be added after word (KHI) or re-organized, as not all of these references (5-8) address or quantify the plasma transport related to KHI but discuss various other aspects of the KHI. For example, add references 5,6,8 after word "(KHI)", then references 25, 7 and at least few for each category (i-iii) of the following missing ones after the word "considerable transport":

iKinetic scale mixing in association with KHI:

Fujimoto and Terasawa, 1991, 1994, 1995

Thomas and Winske 1993

Huba 1996

Matsumoto and Hoshino 2006b

Cowee et al. 2009
Delamere et al, 2011
Nakamura and Daugton, 2014

- ii) Blobby transport in association with 2-D and 3-D reconnection in KH vortices:
Ref. 25 and Nykyri and Otto, 2004 (2-D MHD and Hall-MHD)
Ma et al. 2014a,b (3-D KHI and reconnection, with either KHI or reconnection as primary process)
- iii) Transport via coupling through KHI and kinetic scale waves, e.g Kinetic Alfvén waves:
Chaston et al., 2007;
Johnson, J. R., C. Z. Cheng, and P. Song (1997).

Following the reviewer's suggestions, we changed the references in these sentences at line 33. We here newly added Matsumoto and Hoshino (2006), Chaston et al. (2007), Cowee et al. (2010) and Ma et al. (2014) as references in addition to Nykyri and Otto (2001) and Nakamura and Daugton (2014) to cover all categories the reviewer indicated, each of which predicts a different "considerable transport" mechanism.

2.

Lines 31-46: The discussion related to KH driven plasma transport is rather short and does not differentiate well between

- i). Macroscopic "blobby" transport (Type B) due to KH driven reconnection (when magnetic field components are initially parallel across shear flow layer, and where KHI as a primary mechanism generates anti-parallel components leading to reconnection and between
- ii). The Type A (This present MMS case and Nykyri et al. 2006, angeo Cluster case) where magnetic fields were initially anti-parallel between magnetosheath and magnetospheric field, where the twisting of the boundary by KHI enables reconnection, because flow becomes sub-Alfvénic. I believe this Type A aspect was first clearly addressed in the following simulation paper: Chen, Q., A. Otto, and L. C. Lee (1997). Tearing instability, Kelvin-Helmholtz instability, and magnetic reconnection. *J. Geophys. Res.* 102, 151–161 and should be referenced.

We revised the description related to this topic in lines 41-48. In this paper, we focus on the realistic magnetopause conditions obtained from the MMS spacecraft, which feature both the velocity shear and the magnetic shear (anti-parallel field component). As the reviewer as well as past linear theories suggested, in such realistic conditions, "Type A" vortex-induced reconnection (VIR) would be a dominant process to cause efficient plasma transport along the reconnected field lines. On the other hand, when the magnetic shear is zero or sufficiently weak, the Type-B VIR can also be driven within the highly rolled-up vortex. However, since such an extreme case is not the main focus of this paper, we emphasized that we refer to Type-A VIR as simply VIR, in order to avoid making the paper longer. We here newly cited Nakamura et al. (2006) and (2008) for the linear theories and Nykyri et al. (2006) and Chen, Otto and Lee (1997) for previous studies of the Type-A.

3.

Line 69: How is "Turbulent development" determined. Was there a specific turbulent power law that was observed in the simulations? What is the detailed physical mechanism for jet decay, e.g did the jets excite some secondary instability or kinetic wave modes allowing for the secondary mixing to happen?

Yes, we can see clear turbulent power laws as also mentioned in the above response on the “Title”. The past 3D fully kinetic simulations of guide-field reconnection in the asymmetric layer (Daughton et al., 2014) demonstrated that when considering a strong guide field, the ion-scale reconnection layer with ion-scale jets is significantly disturbed by the secondary 3D tearing mode, leading to the formation of a turbulent reconnection layer. In their simulation, the magnetic field power spectrum of this turbulent layer shows a clear power law with index $-8/3$. The present simulation during the jet decay process (shown in the new Fig.2b) also shows a similar power law in the magnetic field spectrum. This indicates that a similar formation process of the turbulent reconnection layer and a resulting decay process of the ion-scale jet structures also occur in the present simulation. We described on this topic in line 73-89.

4. Lines 71-81 and Figure 1 & 4: I understand that the mass flux across the mixing surfaces j_m is computed using $j_m \sim n_e n_0 V_0$, where V_0 is the velocity shear of 355 km/s. And the large value for the j_m is obtained because the ion jets grow mainly along L-direction, with $U_{iL} = 0.76 V_0$. However, the U_{iN} would be much smaller and would therefore give more modest transport rates if transport from magnetosheath into magnetosphere along N-direction was considered. Based on the Figure 1b, it looks like the width of the transport layer along N direction is only about $(3-4)d_i$, so while efficient mixing along L is occurring it is initially localized to quite narrow area. In Figure 4 it is clear after $t_a = 7.95$ that a larger area gets affected with a width (along N) comparable to the amplitude of the fastest growing KH mode. However, I think the $U_{iL} = 0.76 V_0$ value used, possibly overestimates the transport and perhaps a better way to estimate the maximum transport at $t_a = 7.95$ from magnetosheath to the magnetosphere would be to use following “entry”-velocity $V = (n_e m_e \text{Area}) / (\rho_{\text{sheath}} t \lambda)$, and the diffusion coefficient would be then given by multiplying this velocity by the thickness of the velocity shear layer or the amplitude of the large-scale KH mode. Looking at Figure 4 at $t_a = 6.42$, it is obvious that if this simulation was generated for the conditions with initially parallel-B-fields across shear flow layer (Type B reconnection in KH vortices), the reconnection would occur in high plasma density current layer resulting in “blobby” transport of two larger scale magnetic islands giving roughly same order of value of $V = (n_e m_e \text{Area}) / (\rho_{\text{sheath}} t \lambda)$ as estimated here by eye for present results. This issue should be clarified in the revised version.

We have not computed the mass flux using the flow velocity components, but computed by using a similar way to the “entry velocity” the reviewer suggested. We have computed the mass flux shown in this paper by directly computing the time variation of the integrated mass entering the mixing region across the mixing surfaces as $j_m = (d\Sigma m / dt) / (L_x L_z)$. Thus, we believe that the time evolution of the mass flux shown in this paper (Fig. 5c) directly quantifies how efficiently the magnetospheric and magnetosheath particles enter the mixing region. We modified the description of the definition of the mass flux in the caption of Fig. 5 in line 463-464.

Regarding the Type-B vortex-induced reconnection, we see no evidence of the Type-B VIR in the present simulation, as is the case in the smaller-scale kinetic simulation in Ref. 22. As discussed in Ref.22, this could be because the primary (Type-A) and secondary tearing mode relaxes the stressed current layers formed by the wrapped-up field lines. Since this point has already been described in Ref. 22, we decided not to include it in this paper.

5. Lines 127-130: Discuss here or in the beginning (see comment 1.) more thoroughly the literature related to plasma transport calculations: what methods and plasma approximations were used and what rates were obtained using MHD, Hall-MHD, hybrid, full particle etc. List the relevant missing

references: e.g, Nykyri and Otto, 2004, Angeo (Hall-MHD), Cowee et al, 2009, 2010 (hybrid), etc.
....

We think that describing how each transport quantity (R_{entry} or D) shown in each past referenced paper was computed is not feasible in the allowed space for this letter. Instead, we briefly contrast our new predictions with a few previous estimates in lines 147-157 as follows; $R_{\text{entry}} \sim 10^{24} - 10^{27} \text{ s}^{-1}$ which were estimated from in-situ observations, and $D \sim 10^9 - 10^{10} \text{ m}^2 \text{ s}^{-1}$ which were estimated from in-situ observations and various methods of simulations. To cover the past predictions for the KHI more thoroughly, we also added references to Matsumoto and Hoshino (2006), Chaston et al. (2007) and Cowee et al. (2010).

Regarding the difference depending on simulation methods, although there have been a number of simulation studies of the KHI using single (i.e., MHD) and multi-fluid (i.e., Hall-MHD and two-fluid) and hybrid kinetic-ion/electron-fluid simulations, the fully kinetic and 3D treatments are required to reproduce the precise physics of VIR, since this is the only approach that permits the entire spectrum of secondary tearing modes to develop (cf., Nakamura et al., 2013), and truly quantify plasma transport (cf., Daughton et al., 2014). In other words, the 3D fully kinetic modeling is required to quantitatively discuss the transport quantities related to the (type-A) VIR. We added the description on this point in lines 193-200. In addition, we have added Supplementary Figs. 2a and 2b, which compare the present large-scale 3D fully kinetic simulation and the past smaller-scale 3D fully kinetic simulation (Nakamura et al., 2013). Interestingly, this comparison predicted that the formation and quick decay of ion-scale VIR jet structures would occur when the spatial and growth time-scale of the KHI are much larger than the ion-scales ($\lambda_{KH} \gg d_i, \alpha^{-1} \gg \Omega_i^{-1}$). These conditions are commonly satisfied for normal events of the KH waves observed at the Earth's magnetopause including the MMS event analyzed in this paper. The present large-scale simulation is the first one that satisfies these conditions, thus allowing us to quantitatively investigate the ion-scale VIR physics and the resulting efficient plasma transport. Additional discussion on this topic has been added at lines 159-207. We include new references to Chen et al. (1997), Nykyri and Otto (2001; 2004), Knoll and Chacon (2002), Nakamura et al. (2006; 2008) for past single and multi-fluid simulations, and Cowee et al. (2010) and Delamere et al. (2011) as past hybrid simulations.

6. Lines: 157-158: Please denote, the value of the electron and ion plasma beta in the magnetosheath and magnetosphere, as this can have an effect on what kind of kinetic plasma wave modes could be excited into the system.

Following the reviewer, we added the description of the initial plasma beta in the magnetosheath and magnetosphere in line 232.

7. Line 167: electron to ion mass ratio of 25 is used. Address, how would a more realistic mass ratio affect the results? For example, either by making one simulation with higher or lower mass ratio than for the presented case and discussing the results. I presume that for realistic mass ratio the electrons would stay magnetized longer, and in order to resolve the electron-scale kinetic wave-modes that could contribute to electron de-magnetization, a very high resolution run would be required to resolve the electron-scale wave length waves. Here the numerical resolution of 7 km is appropriate for mass ratio of 25, but probably insufficient if addressing a mass ratio of 1836. Include in methods some discussion related to electron de-magnetization physics, numerical resolution and mass ratios. What terms/physics mostly contribute to the breaking of the frozen in condition for electrons in these simulations (e.g plasma wave modes producing anomalous resistivity, electron inertia etc., off-diagonal components of electron pressure tensor). If you think the details of electron

reconnection physics are not important for ion-scale jet generation and subsequent transport, it would be good to explicitly state and discuss this briefly.

As suggested by the reviewer, since the initial source of both the primary jet and secondary turbulence is the electron-scale tearing modes, these electron-scale modes are affected by the ion-to-electron mass ratio m_i/m_e . Since the guide-field in the present run is rather strong ($\sim 5\text{-}10\times$ of the in-plane anti-parallel component), all electrons are well magnetized, and the corresponding gyro-radius ρ_e for the electrons is a few times smaller than the electron inertial length d_e . This ordering ($d_e > \rho_e$) does not depend on m_i/m_e (but only the electron beta, which is physically matched to MMS event). In addition, the initial spatial size of the primary and secondary tearing modes scales with $d_e \propto (m_i/m_e)^{-0.5}$ depending on the initial thickness ($\sim d_e$) of the compressed current layers, as shown in Nakamura et al. (2013; 2016). Thus, the initial spectrum of tearing modes scale as $(m_i/m_e)^{-0.5}$. However, past kinetic studies suggested that electron-scale tearing modes are rapidly merged into ion-scales within a short time ($\Delta t < \text{a few } \Omega_i^{-1}$) (Karimabadi et al., 2011 for the primary tearing mode; Nakamura et al., 2016 for secondary 3D tearing mode). Thus, at realistic mass ratio ($m_i/m_e = 1836$) one would expect a broader spectrum of tearing modes that rapidly merge towards larger ion-scale structures, resulting in similar overall transport. As also suggested by the reviewer, the formation and decay processes of the ion-scale jets, which develops on much larger spatial and time-scales than ion-scales, should not be significantly affected by the initial electron-scale modes. Although additional 3D runs are not feasible due to the limit of our computer resources, in the revised draft we have performed two additional 2D runs with the same settings in the x-y plane as the main 3D run for $m_i/m_e = 25$ and $m_i/m_e = 9$. In these 2D cases, although the secondary 3D tearing mode is not expected to occur, the primary ion-scale reconnection jets with no oblique angles are reproduced. As shown in Supplementary Fig. 2c, the mass flux for these two runs do not significantly depend on m_i/m_e . We added the description related to this topic in 182-187.

8. Magnetic field vectors are missing in Figure 1b (current density plot), as adding these would make this plot difficult to read, perhaps make one of the figures larger (or put into supplementary information section) and draw the b-field vectors into it. This makes it easier for the reader to understand where the largest shear occurs at the initial system and how the shear evolves.

The previous version of the in-plane magnetic field lines in Fig.1b are too thin to visibly identify the lines. We changed the width of the lines to show them more clearly.

9. Line 154-156: This method to estimate the rotation angle to the direction of the k-vector of the fastest growing mode has been used in 3-D KH simulation studies which was created after Cluster KHI event (Nykyri et al., 2006, angeo), with similar magnetic field geometry across shear flow plane as for the presented MMS event, and these should be referenced here: Adamson E., Nykyri K., Otto A., The Kelvin-Helmholtz Instability Under Parker-Spiral Interplanetary Magnetic ... 218-230, doi:10.1016/j.asr.2015.09.013., 2016.

We newly cited the suggested paper in line 221.

Reviewer #3:

Report on “Turbulent mass transfer caused by vortex-induced reconnection in collisionless magnetospheric plasmas” by Nakamura et al.

The authors performed one new larger-scale kinetic simulation to investigate the evolution of Kelvin-

Helmholtz instability (KHI) and presented ion-scale jets from the vortex-induced reconnection (VIR) rapidly decay through self-generated turbulent reconnection. The authors also compared with in-situ observations by MMS mission. In addition, the authors estimated mass transfer rate nearly one-order higher than previous expectations for the KHI. This manuscript gives some new results that can help to understand how solar wind enters the planetary magnetosphere. However, except the large-scale simulation gives some features of ion-scale jet and the higher mass transfer rate, the author did not present novel findings to meet the standards for publication in Nature Communication. Thus, I think the manuscript may not be suitable for publication in Nature Communication. Below I will give some main comments.

We sincerely appreciate the reviewer's understanding and useful comments, which are variable and helpful for improving our manuscript. To highlight novel findings of this paper, we newly added figures for the turbulent spectrum (Fig. 2) and the comparison with past smaller simulations (Supplementary Figs. 1 and 2) and revised the corresponding sentences in the text as follows.

1. The authors claimed that mass transfer rate is nearly one-order higher than previous simulations. Compared this simulation with the authors' previous simulations [Ref 14], only difference is the box size. Why mass transfer rate increases one order with the increase of the box size? That's so confuse! The authors should give some physical explanations.

As the reviewer suggested, a main difference between the present 3D fully kinetic simulation and the past 3D fully kinetic simulations in Ref. 13 (Nakamura and Daughton, 2014) and Ref. 22 (Nakamura et al., 2013) is the system size of the simulations ($\lambda_{KH}=100d_i$ v.s. $\lambda_{KH}=15-30d_i$). As shown in Supplementary Fig. 2b, the diffusion coefficient of the smaller-scale simulation in Ref. 22 in the early non-linear growth phase of the KHI ($t\alpha < 6$) is more than one-order smaller than the present simulation. This is because (i) the ion-scale jets from the primary reconnection do not appear (i.e., the reconnection does not mature) and (ii) the secondary 3D turbulent reconnection does not occur in the smaller simulation as shown in Supplementary Fig. 1b. The past kinetic simulation results showed that the distance between the primary X-lines scales with λ_{KH} and is of the order of $0.1\lambda_{KH}$, and that the decay time-scale of the primary jet structures by the secondary 3D turbulent reconnection is $\Delta t \sim 30-50\Omega_i^{-1}$. Based on these results, the conditions for the ion-scale jet formation and the decay of the jets within the early non-linear phase (the time between $t\alpha \sim 4.5-6$) can be predicted as $\lambda_{KH} > 50d_i$, $\alpha^{-1} > 50\Omega_i^{-1}$ (i.e., the spatial and growth time-scale of the KHI are much larger than the ion-scales). The present large-scale simulation ($\lambda_{KH}=100d_i$, $\alpha^{-1}=\lambda_{KH}/V_0 \sim 90\Omega_i^{-1}$) and the MMS event analyzed in this paper ($\lambda_{KH} \sim 300d_i$, $\alpha^{-1} \sim 300\Omega_i^{-1}$) easily satisfy these conditions, while the past smaller-scale kinetic simulations ($\lambda_{KH}=15-30d_i$, $\alpha^{-1} \sim 11-21\Omega_i^{-1}$) do not satisfy them. Thus, the present large-scale 3D fully kinetic simulation allowed us for the first time to quantitatively investigate the ion-scale VIR physics and the resulting very efficient plasma transport. We largely revised the text on this topic in lines 159-207.

2. The authors didn't propose any new mechanisms for the mass transfer from solar wind into the magnetosphere. Their manuscript still investigates Kelvin-Helmholtz instability which is well-known to play crucial role on the mass transport across the magnetopause.

As mentioned in our reply to the above reviewer #3's comment-1, the efficient mass transfer rate, which is much larger than the past predictions for the KHI ($D \sim 10^{11} \text{ m}^2 \text{ s}^{-1}$ in the present simulation, while $D \sim 10^9-10^{10} \text{ m}^2 \text{ s}^{-1}$ in the past predictions), results from the formation and quick decay of the

ion-scale reconnection jets, which are newly observed in this paper. In addition, the rapid variations of plasma moments produced by the ion-scale jets seen in Fig.4b can only be resolved using the new high-resolution MMS measurements. Taken together, the present simulation and observational results suggest that the mass transfer from the VIR may be significantly stronger than previously thought. We believe that these new findings are crucial not only for understanding how solar wind enters the Earth's magnetosphere but probably also for understanding mass transfer systems at similar outer boundaries at various magnetospheres and the heliosphere. We revised the text to highlight these points in lines 189-207.

3. The author emphasized the important role of turbulent mass transfer caused by vortex-induced reconnection. However, they didn't show the evidence of self-generated turbulent reconnection, and didn't explain what's turbulent mass transfer and what's the difference with previous mass transfer mechanism.

We added a new figure (Fig. 2) which shows the turbulent spectra. As we also described in our reply to the reviewer #1's comment-3, the past 3D fully kinetic simulations of guide-field reconnection in the asymmetric layer (Daughton et al., 2014) demonstrated that when considering a strong guide field the ion-scale reconnection layer with ion-scale jets is significantly disturbed by the secondary 3D tearing mode, leading to the formation of a turbulent reconnection layer. In their simulation, the magnetic field power spectrum of this turbulent layer shows a clear power law with index $-8/3$. The present simulation during the jet decay process (shown in the new Fig.2b) also shows a similar power law in the magnetic field spectrum. This indicates that a similar formation process of the turbulent reconnection layer and a resulting decay process of the ion-scale jet structures also occur in the present simulation. We described on this topic in lines 73-89.

In addition, as mentioned in our reply to the above reviewer #3's comment-1, since the time-scale of the turbulent decay of the primary jets is $\Delta t \sim 30-50\Omega_i^{-1}$, this decay process can occur within the early non-linear growth phase of the KHI ($t\alpha \sim 4.5-6$ where $\alpha \sim 90\Omega_i^{-1}$) in the present simulation. Correspondingly, the efficient mass transfer related to the secondary turbulence occurs in the early non-linear phase in the present large-scale simulation, while the turbulent feature can be seen only in the late non-linear phase ($t\alpha > 7$) in the past smaller-scale simulations. We believe that this turbulent induced mass transfer during the growth time scale ($\alpha^{-1} = \lambda_{KH}/V_0$) of the KHI is an important new finding of this paper. We described on this topic in lines 171-182.

Reviewers' comments:

Reviewer #2 (Remarks to the Author):

The authors have clearly addressed all the comments raised by this reviewer. I think this is an excellent paper now and suitable for publication in Nature Communications.

This reviewer does not need to see this paper again before submission.

Very minor things to be fixed for final submission:

Line 54: extra space after y : (x, y, z)

Line 168: add a space after $KH =$

Supplementary figures line 35: extra space between "(" and "but" and extra space before "Although".

Reviewer #3 (Remarks to the Author):

The authors have largely improved the manuscript from the previous version, and also addressed my concerned comments, especially highlighted the novel findings of the paper. I think this paper may be suitable to publish in Nature Communications after some revisions.

The authors have shown the turbulent spectra and compared with the simulation results of Daughton et al. [2014]. Actually I suggest the authors to compare with the turbulent spectra from in-situ satellite observations in the reconnection region or jet (such as Eastwood et al. 2009 PRL, 102, 035001; Huang et al. 2012 GRL, 39, L11104; Osman et al. 2015 ApJL, 815, L24), in the turbulent plasmas (such as Sahraoui et al. 2006 PRL, 96, 075002; Alexandrova, et al. 2012, ApJ, 760, 121; Sahraoui et al. 2013 ApJ, 777, 15; Huang et al. 2014 ApJL, 789, L28 ; Breuillard, et al. 2016, ApJ, 829, 54; Huang et al. 2017 ApJL, 836, L10). Such comparison would give much stronger evidence for the turbulent evolution of the VIR in their simulations.

I think there are some mistakes in the Supplementary Figure 2 . The legend shows black lines, black dashed lines, and black dashed-dotted lines, but one cannot see them in the Figure 2. Please check this.

RESPONSE TO REFEREES

We would like to thank the reviewers for their careful reading of our revised paper. We have taken the additional comments from the reviewers into account in the revised manuscript. Below are our responses to each comment.

Reviewer #1:

The authors have clearly addressed all the comments raised by this reviewer. I think this is an excellent paper now and suitable for publication in Nature Communications.

This reviewer does not need to see this paper again before submission.

We sincerely appreciate the reviewer's understanding and positive comments.

Very minor things to be fixed for final submission:

Line 54: extra space after y: (x, y, z)

Line 168: add a space after KH =

Supplementary figures line 35: extra space between "(" and "but" and extra space before "Although".

Following the reviewer, we appropriately removed and added space in these sentences.

Reviewer #2:

The authors have largely improved the manuscript from the previous version, and also addressed my concerned comments, especially highlighted the novel findings of the paper. I think this paper may be suitable to publish in Nature Communications after some revisions.

We sincerely appreciate the reviewer's understanding and positive comments.

The authors have shown the turbulent spectra and compared with the simulation results of Daughton et al. [2014]. Actually I suggest the authors to compare with the turbulent spectra from in-situ satellite observations in the reconnection region or jet (such as Eastwood et al. 2009 PRL, 102, 035001; Huang et al. 2012 GRL, 39, L11104; Osman et al. 2015 ApJL, 815, L24), in the turbulent plasmas (such as Sahraoui et al. 2006 PRL, 96, 075002; Alexandrova, et al. 2012, ApJ, 760, 121; Sahraoui et al. 2013 ApJ, 777, 15; Huang et al. 2014 ApJL, 789, L28 ; Breuillard, et al. 2016, ApJ, 829, 54; Huang et al. 2017 ApJL, 836, L10). Such comparison would give much stronger evidence for the turbulent evolution of the VIR in their simulations.

We appreciate the reviewer's useful suggestions. We basically agree with the suggestions. However, since the comparison with other turbulent processes in collisionless plasmas is not the main focus of this paper, in order to avoid making the paper longer, we emphasized by referring the past studies suggested by the reviewer that the results of this paper will be compared with the other processes in the future in lines 207-211.

I think there are some mistakes in the Supplementary Figure 2 . The legend shows black lines, black

dashed lines, and black dashed-dotted lines, but one cannot see them in the Figure 2. Please check this.

Following the reviewer's suggestions, we changed the legends of the Figure 2.